# The genetic landscape of benign thyroid nodules revealed by whole exome and transcriptome sequencing

Lei Ye[1,*], Xiaoyi Zhou[1,*], Fengjiao Huang[1], Weixi Wang[1], Yicheng Qi[1], Heng Xu[2], Yang Shu[2], Liyun Shen[1], Xiaochun Fei[3], Jing Xie[3], Min Cao[1], Yulin Zhou[1], Wei Zhu[1], Shu Wang[1], Guang Ning[1,4] & Weiqing Wang[1]

The genomic alterations for benign thyroid nodule, especially adenomatoid nodule, one of the most common types of hyperplasia lesion, are ill-studied. Here, we show whole-exome sequencing and/or transcriptome sequencing data on adenomatoid nodules with or without coincidental papillary thyroid carcinoma (PTC). Somatic mutation of *BRAF* (22/32) is only detected in PTC, while mutations in *SPOP* (4/38), *ZNF148* (6/38) and *EZH1* (3/38) are found enriched in adenomatoid nodule. In an expanded cohort of adenomatoid nodule ($n=259$) mutually exclusive $SPOP^{P94R}$, $EZH1^{Q571R}$ and *ZNF148* mutations are identified in 24.3% of them. Adenomatoid nodules show very few overlapped mutations and distinct gene expression patterns with their coincidental PTC. Phylogenetic tree analysis uncovers that PTCs evolved independently from their matched benign nodules. Our findings reveal that benign nodules possess a unique molecular signature that differs from PTC and provide genomic evidence for the conventional belief that PTC and benign nodules have independent origin.

[1] Shanghai Key Laboratory for Endocrine Tumors, Shanghai Clinical Center for Endocrine and Metabolic Diseases, Shanghai Institute of Endocrine and Metabolic Diseases and Shanghai E-institute for Endocrinology, Ruijin Hospital, Shanghai Jiao Tong University, School of Medicine, 197 Ruijin 2nd Road, Shanghai 200025, China. [2] Department of Laboratory Medicine, Precision Medicine Center, State Key Laboratory of Biotherapy and Precision Medicine Key Laboratory of Sichuan Province, West China Hospital, Sichuan University and Collaborative Innovation Center, Chengdu 610041, China. [3] Department of Pathology, Ruijin Hospital, Shanghai Jiao Tong University, School of Medicine, 197 Ruijin 2nd Road, Shanghai 200025, China. [4] Laboratory for Endocrine & Metabolic Diseases of Institute of Health Science, Shanghai Jiao Tong University School of Medicine and Shanghai Institutes for Biological Sciences, Chinese Academy of Sciences, 227 South ChongQing Road, Shanghai 200025, China. * These authors contributed equally to this work. Correspondence and requests for materials should be addressed to W.W. (email: wwq10320@rjh.com.cn).

The routine use of sensitive ultrasound examination has led to a dramatic increase in the prevalence of thyroid nodule detection, with only 5% cases having a clear malignant diagnosis[1,2] Large efforts have been taken to characterize the genetic profiling of papillary thyroid carcinoma (PTC)-the most common malignant thyroid nodules. As a result of The Cancer Genome Atlas (TCGA) project and other studies tumorigenesis related genetic events have been uncovered in 96.5% of PTC (for example, *BRAF* mutation, *RAS* mutation and so on)[3]. However, the genomic characters of benign nodules are much less understood. Recent studies have revealed novel genetic alterations associated with follicular adenoma, the benign neoplastic thyroid nodule[4,5]. While the genetic alterations for benign hyperplasia thyroid nodule, especially adenomatoid nodule, one of the most common hyperplasia lesions and undistinguished from follicular thyroid carcinoma in fine need aspiration[6], are ill-studied. In this study, we found 24.3% of adenomatoid nodule carried mutually exclusive $SPOP^{P94R}$, $EZH1^{Q571R}$ and *ZNF148* mutations, a unique molecular signature that differs from PTC. We also provide genomic evidence for the conventional belief that PTC and benign nodules have independent origin.

## Results

### Benign nodule harboured unique variations distinct from PTC.

To gain insight into the genetic mechanism of thyroid nodule development, we examined 127 DNA and 40 RNA samples, from 21 patients with coincident benign nodule and PTC (TB patients), and eight patients with benign nodule burden only (SB patients) by whole-exome and transcriptome sequencing (for sampling details, see Supplementary Data 1, for clinical characteristics of patients, see Supplementary Data 2). The coincident benign nodule and PTC also gave us an opportunity to provide genomic evidence for the conventional belief that benign nodule and PTC has independent origin[7]. To be noted, all the benign nodules analysed in the present study are adenomatoid nodule.

Matched tissues were taken from each TB patient (blood, benign nodule, PTC and normal thyroid) and SB patient (blood, benign nodule and normal thyroid) (Fig. 1a). Diagnosis of PTCs were made by criteria defined by the World Health Organization[8]. DNA obtained from these tissues was submitted to whole-exome sequencing with an average effective coverage of $161\times$ (ranging from 130 to 180) (Supplementary Data 3). In total 688 somatic mutations targeting 528 genes were identified in normal thyroids, benign nodules and PTCs using matched blood DNA as germline references (Supplementary Data 4). PTCs had a significantly higher mutation rate (0.33 mutations per Mb) than normal thyroid (0.12 mutations per Mb) (Wilcoxon signed-rank test, $P = 8.6 \times 10^{-5}$), but a similar rate with benign nodule in either TB (0.34 mutations per Mb) or SB patients (0.38 mutations per Mb) (Fig. 1b, Supplementary Fig. 1). Despite the similar mutation rate of benign nodules and PTCs, their genetic profiles were clearly distinct (Supplementary Fig. 2). Consistent with the finding of TCGA project[3], we found our PTC tumours harboured a low mutation rate compared to other cancer types, and highly mutated in *BRAF* gene (16/20 patients, 80%) mainly at its 600 residue ($BRAF^{V600E}$) (Fig. 2 and Supplementary Data 4). Additional PTC-specific mutations were found to co-exist with $BRAF^{V600E}$ (for example, *FAT3*); however, for four patients no driver mutations were observed (Fig. 2 and Supplementary Fig. 3). The mutation landscape for benign nodules was completely different from PTCs. We observed mutually exclusive mutation of *ZNF148* (21.4%), *SPOP* (14.3%) and *EZH1* (10.7%) in benign nodules, but not PTC or normal thyroid. *ZNF148* mutations were either nonsense or frameshift, all

located in the last exon. When DNA sequence was performed on an additional 231 benign nodules, there was a drop in the overall frequency of *ZNF148* mutation (5.4%), while $SPOP^{P94R}$ (11.2%) and $EZH1^{Q571R}$ (9.3%) mutation frequencies remained similar and continued to be mutually exclusive (Supplementary Fig. 4). In TCGA database, the highest mutation frequency for *SPOP* was 7.63% (38/498) in prostate cancer, *ZNF148* 11.25% (9/80) in uveal melanoma 14.3%) and *EZH1* 11.25% (9/80) in uveal melanoma (Supplementary Fig. 5 and Supplementary Table 1). Only $EZH1^{Q571R}$ showed slightly increased cell proliferation ($P = 0.04$) and reduced cell invasion ($P = 0.04$) when compared to WT in the functional study using normal human thyroid cell line (Nthy) (Supplementary Fig. 6). Beside SNV and indels, we also analysed copy number variation (CNV) in tumour and benign samples. We found a previous reported PTC-related arm-level alternation, 22q-del, in nine tumour samples (from seven patients), but not in any benign sample (Supplementary Fig. 7). No recurrent arm-level CNVs was found in benign samples. Moreover, we did not detect any recurrent or functional transcript fusion in either PTC or benign nodules (Supplementary Fig. 8).

### Little mutational overlap exists between paired PTC and benign nodule.

To compare the genetic characteristics of benign nodules with thyroid cancer, we performed paired comparison in each TB patient. Patients with multiple non-contiguous benign nodules or PTC foci (multi-foci) or multiple geographically distinct areas from the same foci (multi-areas) served as positive controls. As expected, mutations identified in distinct areas within a single PTC foci exhibited a higher level of overlap (from 14.3 to 62.5%, median = 40%), compared to independent PTC foci from the same patient (9.1 and 25%) (Supplementary Fig. 9). PTC and benign nodule had the lowest frequency of overlapping mutations (from 0 to 21.4%, median = 7.7%). Moreover, when considering the matched normal tissue, only 9 PTC and benign nodule pairs exhibited overlapping mutations (from 1.7 to 7.4%)(Fig. 3). The low overlap ratio appeared not caused by sample purity because the mutation number was not significantly correlated with the purity of the samples (Spearman's rho, $r = -0.037$, $P = 0.76$) (Supplementary Data 5). We compared the variation allele fraction (VAF) of mutations in paired samples for each patient (Supplementary Figs 10 and 15–32), and found that the PTC-benign nodule correlation was always the lowest among the sample pairs of each patient (Supplementary Fig. 11).

### The gene expression in benign nodules differs from PTC tumors.

We further explored gene expression patterns by transcriptome sequencing in a subset of 26 available samples derived from nine patients, including five TB and four SB patients (set 1 in Supplementary Data 1 and 6). We evaluated 21,441 transcribed genes (average FPKM mapped reads $\geq 1$) looking for genes that were differentially expressed over 1.5-fold compared with normal thyroid. We identified both PTC-effective genes (363 up-regulated and 32 down-regulated, including *FN1*, *SERPINA1*, *CDH3*, *TFF3* that have been reported previously, Supplementary Data 7) and benign nodule-effective genes (640 up-regulated and 262 down-regulated, Supplementary Data 8). However, only 8% of up-regulated and 3.1% of down-regulated genes overlapped between PTC and benign nodule (Fig. 4a). Comparison of paired PTC and benign nodule transcriptomes identified 1,969 differentially expressed genes (Supplementary Data 9) and the majority overlapped with PTC-effective genes, with 77% for up-regulated and 81% for down-regulated gene, respectively (Fig. 4b). Hierarchical clustering and PCA clearly separated malignant samples from benign nodule and normal tissue (Fig. 4c, Supplementary

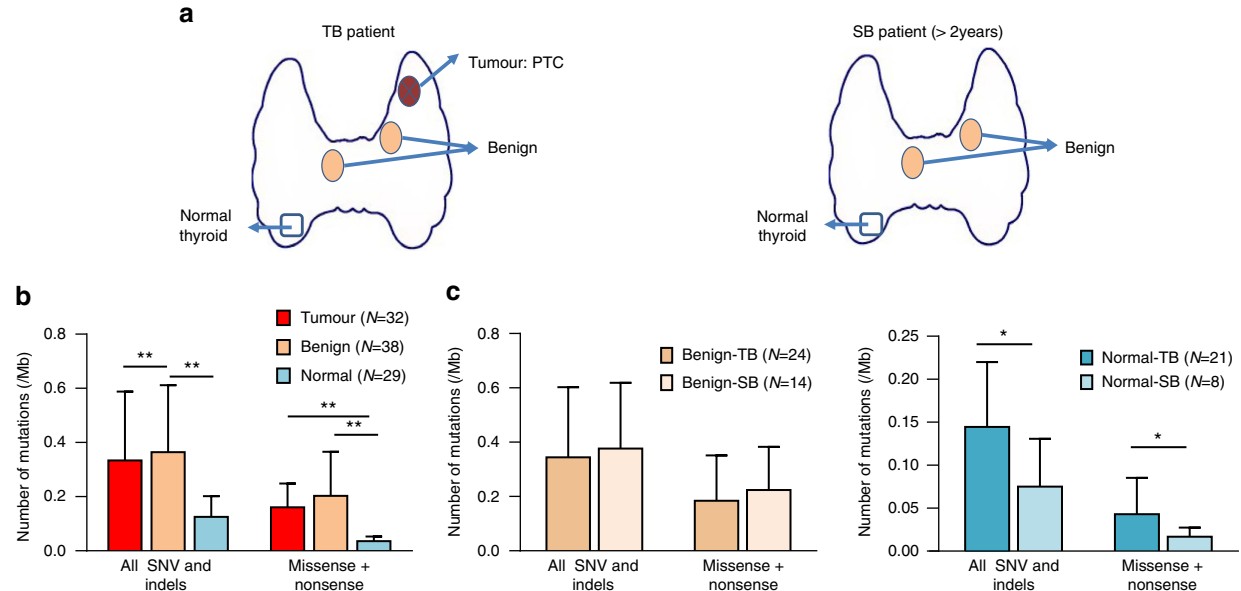

**Figure 1 | Overview of somatic mutations in thyroid tissues.** (**a**) Sampling schematic: Tissue samples were obtained from the thyroids of patients with or without PTC (>2 years follow-up). Sample Groups TB: benign nodule with concurrent PTC; SB: simple benign nodule without concurrent PTC. Normal tissues were at least 2 cm distant from the lesion foci. (**b**) Comparison of number of somatic mutations in PTCs, benign nodule and normal thyroid. Non-synonymous mutations included missense, stopgain SNV and indels. Mean ± s.d., **P < 0.01 Wilcoxon signed-rank test. (**c**) Comparison of number of somatic mutations in benign nodules from TB and SB patients, as well as normal thyroid from TB and SB patients. Mean ± s.d., *P < 0.05 by Wilcoxon signed-rank test.

Fig. 12). Notably, benign nodules from TB and SB patients were clustered together, indicating their functional similarity and distinction from PTC. This finding was validated in an additional 14 samples from four patients (two TB and two SB, set 2 in supplementary Data 1) performed separately with set 1 with the same gene list in Fig. 4a. The only exception was benign nodule foci 2 of PB5 ($EZH1^{Q571R}$), which was clustered with PTCs, consistent with its suspicious tumour feature in pathological examination (Supplementary Fig. 13). This distinct separation of benign nodules and PTC was uniformly found in three established classifiers: the thyroid differentiation score (TDS, 16 genes set related to thyroid metabolism and functions[3]), the ERK-activity score (EAS, 52 genes set for MEK/ERK activation measurement[9]), and the BRAF-RAS signature (71 genes set related to distinguishes $BRAF^{V600E}$ mutation from $RAS$ mutation)[3,9]. Significant higher TDS values were detected in benign nodules (0.01–1.03) than PTCs (−3.4 to −1) (Wilcoxon signed-rank test, $P = 0.0007$), but not normal thyroid. The EAS was extremely high in $BRAF^{V600E}$ positive PTCs (39.2 on average), intermediate in $BRAF^{WT}$ PTCs (0.1 on average), and low in normal tissues (−17.8 on average) (Fig. 4d). Benign nodule showed relatively low but variable scores (16.7 in SB patient and −6.8 in TB patient) (Fig. 4d). For the BRAF-RAS signature, PTCs and benign nodules also segregated in independent branches with the only exception of PB5 (Supplementary Fig. 14).

**Independent evolution of benign nodules and PTCs.** Finally, we constructed phylogenetic trees for all 20 patients in order to illustrate the evolutionary relationships between PTCs, benign nodules and normal thyroid tissues (Fig. 5 and Supplementary Figs 15–32). The majority of patients (15 of 20) exhibited early branched evolution between benign nodules and PTCs, particularly when taking normal thyroid into account (Supplementary Table 2). Branching typically was driven by mutation of common genes (for example, $EZH1^{Q571R}$ in benign nodule and $BRAF^{V600}$

in PTC of PA15; $ZNF148^{S589X}$ in benign nodule of PA12, Fig. 5a–c and Supplementary Fig. 25), suggesting an early separation and distinct evolution patterns between PTC and benign nodule. Five patients uniquely shared phylogenetic branches (PA1, PA6, PA7, PA8 and PA19) between PTC and benign nodule, however all were non-coding variants except in PA8 where the benign nodule and PTC shared an $AIM1L$ mutation (p.531–548del) predicted as disease causing by MutationTaster but found in neither COSMIC nor TCGA[3] (Fig. 5b–d).

**Discussion**
Our studies identified a common characteristic genetic profile of benign nodules, mutually exclusive mutation of $SPOP$, $ZNF148$ and $EZH1$ was observed in 24.3% of the adenomatoid nodules but not in matched PTC tumours. Recurrent $SPOP$ mutation was found in prostate cancer[10]. Of note the P94R mutation, we identified in benign nodules has not been observed before. Although $ZNF148$ has been speculated as a tumour suppressor gene, up-regulated expression was observed at steps towards carcinogenesis in familial adenomatous polyposis[11]. Finally, EZH1 is an H3K27 methyltransferase involved in maintaining stem cell identity and executing pluripotisis[12]. While mutations of the three genes have been identified in TCGA PTCs it is at a very low frequency (1 for $SPOP$, 2 for $ZNF148$ and 2 for $EZH1$ out of 402 sequenced patients with coincidently known PTC driver mutations)[3]. $SPOP$ mutations have been found in TCGA prostate cancer (10.2%), but with ~70% has altered either Tryptophan residue at 131 position or Phenylalanine residue at 133 position (Supplementary Fig. 4), while mutations of $ZNF148$ and $EZH1$ were commonly observed in melanoma as well with no hot mutation spot. However, patients tended to have co-existence of alterations in well-known cancer-related genes (for example, $BRAF$ and $MLL2$) in these malignant tumours, which is distinct from benign nodules. Our findings are more consistent with a causal role on the formation of benign nodules than neoplastic transformation. Indeed, functional studies only found mild

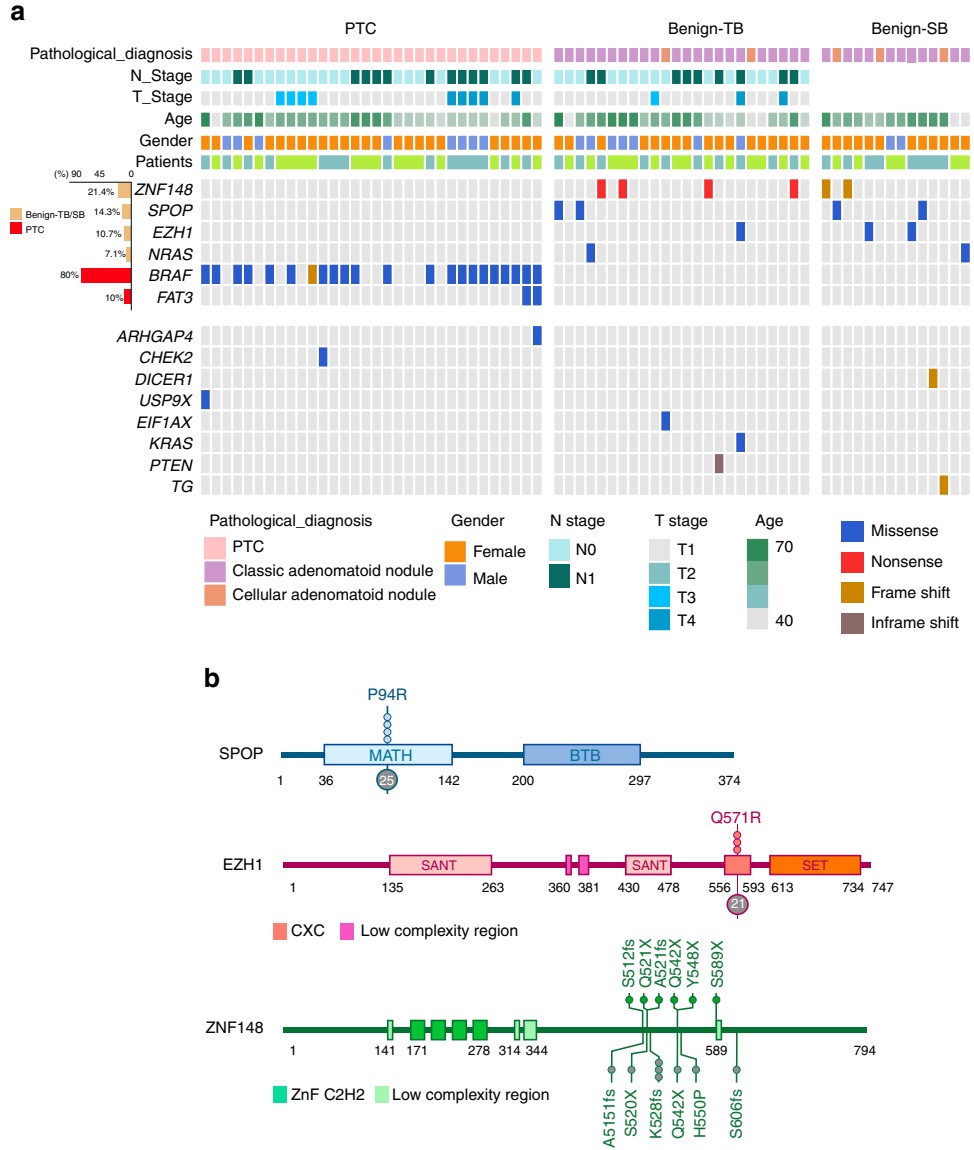

**Figure 2 | Putative functional somatic mutations in thyroid tissues.** (**a**) Landscape of significantly mutated gene and genes previously associated with PTC in thyroid tissues. The frequency of mutation in each sample type was indicated in the left histogram. (**b**) Schematic representation of mutations in SPOP, EZH1 and ZNF148. Conserved domains were indicated as boxes. Somatic mutations detected in our cohort were indicated in the upper. Somatic mutations of these three genes in expanding 231 patients were indicated in grey (bottom).

proliferation and invasion induction of EZH1$^{Q571R}$ in normal thyroid follicular cell line (Supplementary Fig. 14). The formation of thyroid nodular structure in adenomatoid nodule is probably an early stimulus that causes enlargement of the thyroid, such as Iodine deficiency, nutritional goitrogens or autoimmunity[13], followed by local proliferation of follicular epithelial cell due to somatic EZH1 mutations, a genetic path different from thyroid cancer. However, we did not investigate the capacity of thyroid hormone synthesis and transport, which may be compromised in nodules with somatic SPOP, ZNF148 and EZH1 mutation, causing enlarged follicles distended with colloid as featured in adenomatoid nodule. Further studies are needed. Benign thyroid nodule therefore is distinct from cancer precursor lesions, such as melanocytic nevi which harbour recurrent BRAF mutation[14] or benign lesions with risk of malignant transformation, such as usual ductal hyperplasia of the breast, which harbour mutations of the phosphatidylinositol 3-kinase/AKT/mTOR axis[15]. A challenging diagnostic dilemma currently encountered in the clinic is the difficulty differentiating benign follicular lesion from

FTC[2]. We sequenced SPOP$^{P94R}$ and EZH1$^{Q571R}$, and the last exon of ZNF148 in an additional set of 55 follicular thyroid carcinomas (FTC). No mutation was found in FTC. These data suggest these mutations are specific for benign nodule and identify the nodules with these mutations may spare them from surgery when follicular lesion of undetermined significance or a follicular neoplasm was suspected in fine-needle aspiration. Current DNA-based mutation testing of thyroid nodules only includes genes with defined roles in thyroid cancer formation, functioning as 'rule in' tests[16–18]. We believe that the inclusion of SPOP, ZNF148 and EZH1 as benign nodule associated genes, has the potential to increase its diagnostic efficacy of DNA-based mutation testing by providing 'rule out' information.

For many years, we believe PTC does not derive from benign thyroid nodules, mainly from epidemiology data[19–21] and some molecular data, for example, BRAF$^{V600E}$ is in a significant subset of PTC while it is lacking in benign nodule. However, genomic evidence has been lacking. In the present study, we enrolled a unique cohort of patients with concurrent benign nodule and

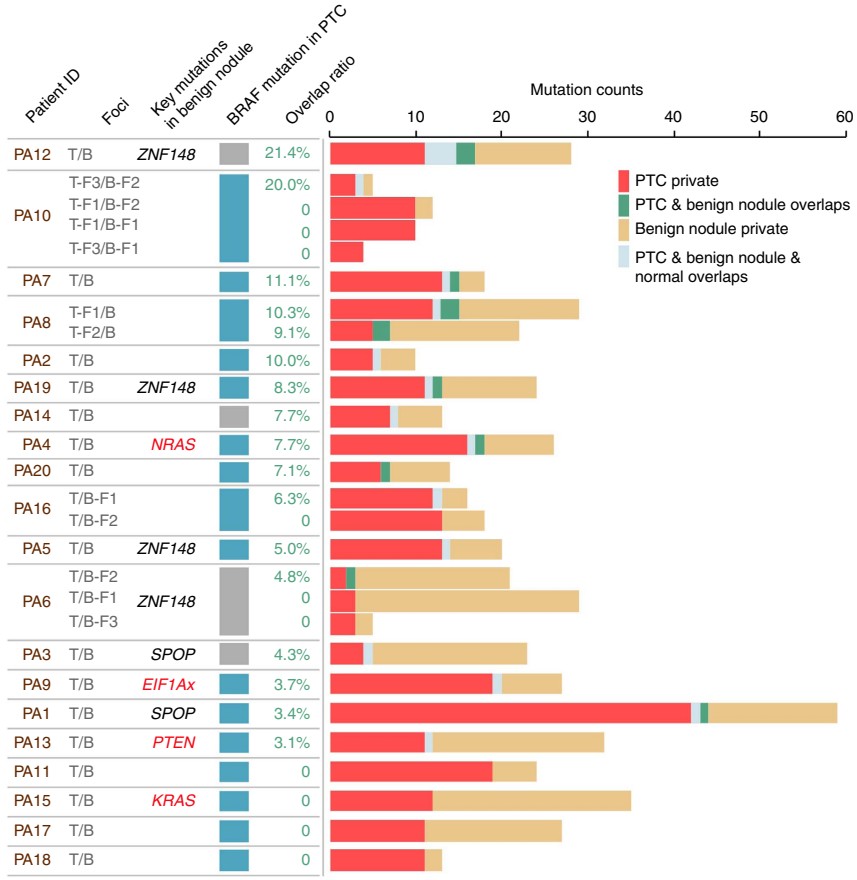

**Figure 3 | Frequency of shared mutations between PTC–benign nodule pairs.** The overall length of each bar represents the total number of mutations for each patient. Patients are ordered by the overlap ratio (indicated in the left of the bar chart).

PTC and systematically investigated the genetic profiles of multi-nodular lesions. In consistent with conventional belief[7], we found benign nodules are genetically unrelated to PTC as define by: few overlapping mutations and distinct branched evolutionary relationships in the paired benign nodule and PTC samples, especially for those PTC with *BRAF* mutation; PTC-related mutations detected in benign nodules did not exist in the paired PTC tumour; and distinct transcriptome clustering and functional pathways between benign nodules and PTCs. Multifocal PTCs have also been demonstrated as independent tumours in most cases by discordant patterns of X-chromosome inactivation[22] and discordant mutational spectra revealed by whole-exome sequencing[23]. In the later study, the authors included one case with concurrent benign follicular adenoma and found completely distinct mutation profiles from the two cancer foci[23], supporting the assertion of independent origins for multiple nodule lesions of thyroid, either benign or malignant.

In conclusion, our data provide strong evidence that benign thyroid nodules have a genetic and transcriptome landscape that is distinct with PTC tumours. These findings confirmed the conventional belief that benign nodule development is distinct from PTC tumour formation. Adding the benign associated mutations in the current molecular diagnostic panel for thyroid cancer may increase its diagnostic efficacy by providing 'rule out' information.

## Methods

**Sampling and DNA extraction.** This study was approved by the Ethics Committee of Ruijin Hospital, School of Medicine, Shanghai Jiao Tong University. All the samples were collected with documented informed consents from the enrolled patients. The tissue samples for sequencing were collected during surgical resection from 29 patients, 21 with coincident benign nodule (adenomatoid nodule) and

PTC (TB patients) and eight patients with benign nodule burden only (SB patients). All patients were treatment naive (chemo- or radiotherapy) at the time of collection. Matched peripheral bloods of the 28 patients (#PA21 excluded) were used as germline reference. SB patients were defined as having at least one nodule for a duration of more than 2 years and no histological indication of malignancy. The tissues were snap frozen in liquid nitrogen and independently analysed to minimize potential contaminations. Hematoxylin and eosin stained tissue sections were examined by an experienced pathologist. Areas with confirmed histological findings were collected for nucleotides extraction (tumour cellularity >80% for PTC). The stained sections were scanned by nanozoomer 2.0-RS (Hamamatsu). In all cases DNA was extracted with QIAGEN DNeasy Blood & Tissue Kit according to the manufacturer (Qiagen, Hilden, Germany).

**Whole-exome sequencing.** Genomic DNA of 127 samples from 28 patients (#PA21 excluded) was randomly sheared through ultra-sonication for about 3 min using Bioruptor XL Sonication System to generate paired-end libraries with an average insert size of ~300 bp. Whole-exome capture was performed using SureSelect Human All Exon 50 Mb kit (Agilent Technologies, Santa Clara, CA), according to the standard protocol. The captured template DNA fragments of the constructed libraries were hybridized to the surface of flow cells and amplified to form clusters, and sequenced using the Illumina HiSeq2500 sequencing system, generating 100-bp paired-end reads. All the samples yielded enough high quality sequencing data with averagely 13.26 Gb raw data after Illumina pass filtering (~160-folds genome coverage, Supplementary Data 3).

**Mutation calling and annotation.** The paired-end reads from whole-exome sequencing were first aligned to human reference genome (hg19) using BWA (version 0.7) with default parameters. PCR duplicates were then marked in the alignments using Picard tools (version 1.1). In the local regions harbouring insertions or deletions, the initial reads were re-aligned and recalibrated for base quality. From the resulting BAM file, the variation calling was carried out by using UnifiedGenotyper in GATK. For the purpose of comparing mutations among matched samples from a single patient, a single-normal-multi-tumour strategy was used based on the GATK joint calling of somatic mutations. To avoid errors related to sequencing or aligning, several strict filtering criteria were applied. We required that both the target tissue and the blood reference should be covered sufficiently (at

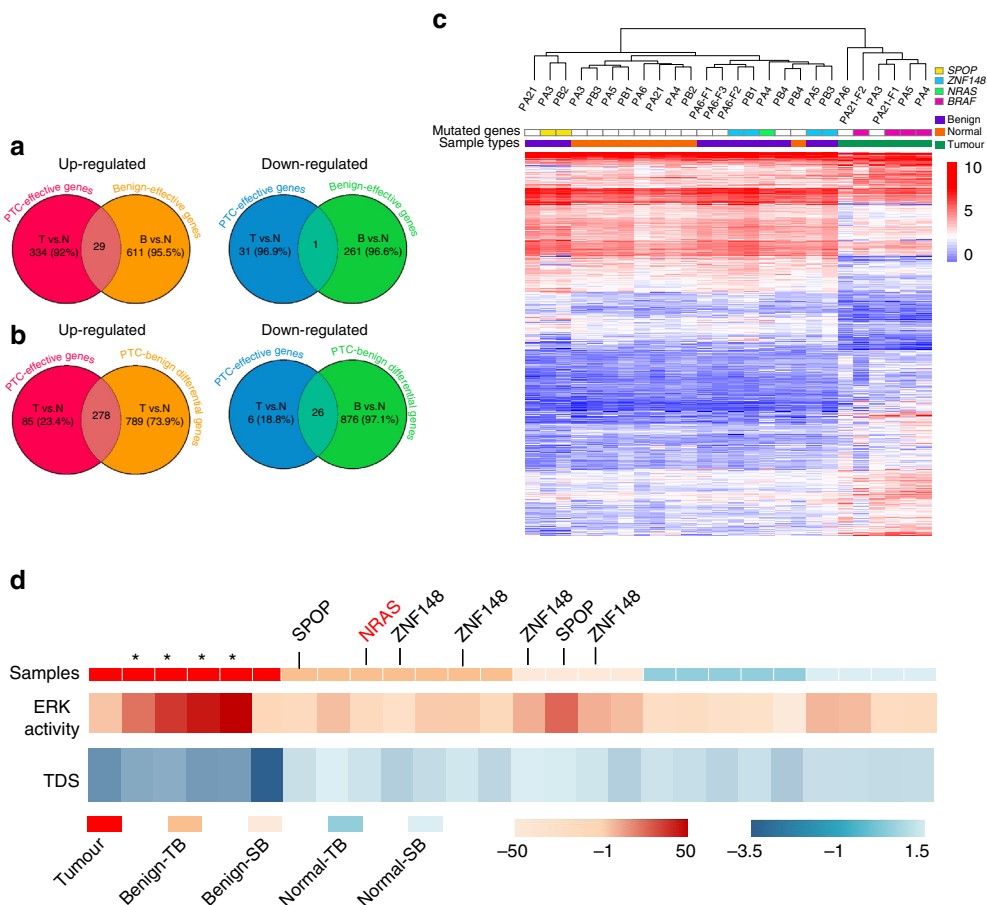

**Figure 4 | Transcriptional patterns defining malignant and benign thyroid tissues.** (**a**) Venn diagrams showing overlaps between PTC-effective and benign nodule-effective genes. T versus N: PTC versus normal tissues (FC > 1.5); B versus N: benign nodules versus normal tissues; (**b**) Overlap of PTC-effective genes (T versus N) with genes differentially expressed between PTC tumour and benign nodules (T versus B). (**c**) Unsupervised hierarchical clustering of transcribed genes (FPKM > 1) and differentially expressed over 1.5-fold for 26 tissue samples. (**d**) TDS and ERK-activity score. Samples were coloured by tissue types (upper panel). The TDS was based on the expression of 16 genes and the ERK activation by the expression of 52 genes. The asterisk indicates the PTC samples with BRAFV600E mutation, and the key mutations in benign nodules were labelled.

least 10 × sequencing depth). Moreover, the mutation should be supported by at least 10% of the local mapping reads in the tissue (set to be 5% if the local depth was > 50), and the mutation should be supported by at least three reads in the tissue. For each candidate somatic mutation site, chi-squared tests were used for the allele depth and frequency of the case and germline samples to eliminate the mutations found in the blood and multiple tissues, and mutations found in more than two reads of the blood germline sample were also removed. The common variants in dbSNP (build 142), the 1,000 genomes (MAF > 5%), the exome aggregation consortium (ExAC MAF > 1%) and the variants in the intergenic region and intronic region were also excluded in the following analysis. The potential effects of the somatic SNV and indels were annotated using ANNOVAR.

**Mutation analysis.** The mutation density was calculated using the assumption that the human exome has 30 Mb in protein-coding genes with sufficient coverage. The base contexts of the somatic mutations were analysed with SNVs as described previously. For the SNVs, there were totally six classes of base substitution (C > A, C > G, C > T, T > A, T > C and T > G). We incorporated the information on the bases immediately 5′ and 3′ to each SNV, and counted the frequencies of each base context. For the analysis of tumour-benign shared mutations, the mutation profiles of paired tumour and benign nodules were compared to derive overlap. Mutations also shared with matched normal tissues are additionally highlighted (Fig. 3). For PA8, PA9, PA12 and PA15, mutations in tumour areas of a same foci were combined together. The relationship of all paired samples was evaluated using the Pearson correlation based on VAF values of somatic mutations between samples, which were displayed in Supplementary Fig. 10 and 11. For each patient, the somatic mutations called from PTC, benign nodules and normal tissues were used to analyse the phylogenetic relationship of multiple samples. A pairwise matrix was calculated from the correlation of VAF values, and then this distance matrix was used to construct unweighted trees using the software PHYLIP (version 3.66)

**Copy number calculation and purity estimation.** Copy number segmentation was performed by RECAPSEG of GATK package. Read depth across the human genome was calculated by the 'CalculateTargetCoverage' module for each sample, and the copy ratios in both tumours and benign nodules were further calibrated using the read depth information in the blood tissues. The copy-ratio profiles were segmented by the 'PerformSegmentation' module. CNV was visualized by plotting the profile data against the human genome using Integrative Genomics Viewer. The ABSOLUTE package (version 1.0.6) was used to estimate the purity and the average ploidy for each tumour sample and each benign nodule sample. We used both the segmented copy-ratio results (the log₂ copy number change) and the allelic depth of all somatic mutations for the estimation of the purity, and further manually reviewed all the ABSOLUTE solutions

**Validation of mutations by PCR and Sanger sequencing.** Exons where the variants of interest located were amplified using primers listed in Supplementary Data 10 on Dual 96-well GeneAmp PCR System 9,700 (Applied Biosystems, Courtaboeuf, France), with 20 ng of template DNA from each sample used per reaction. The products were sequenced by 3730 × l DNA Analyzer (Applied Biosystems, Courtaboeuf, France). All sequences were analysed by the Sequencing Analysis Software (version 5.2, Applied Biosystems, Courtaboeuf, France). All variants of interest were manually confirmed in the resulting trace files.

**Evaluation of recurrent mutations in a larger set of benign nodules.**
A total of 328 fresh-frozen thyroid benign nodule tissues from 259 patients were collected. Genomic DNAs were prepared as described above. The presence of SPOP[P94R] and EZH1[Q571R] mutations was assessed in this cohort by targeted Sanger sequencing of PCR products. For the *ZNF148* gene (NM_021964) PCR products covering all the coding region and flanking intron–exon junctions were subjected to Sanger sequencing.

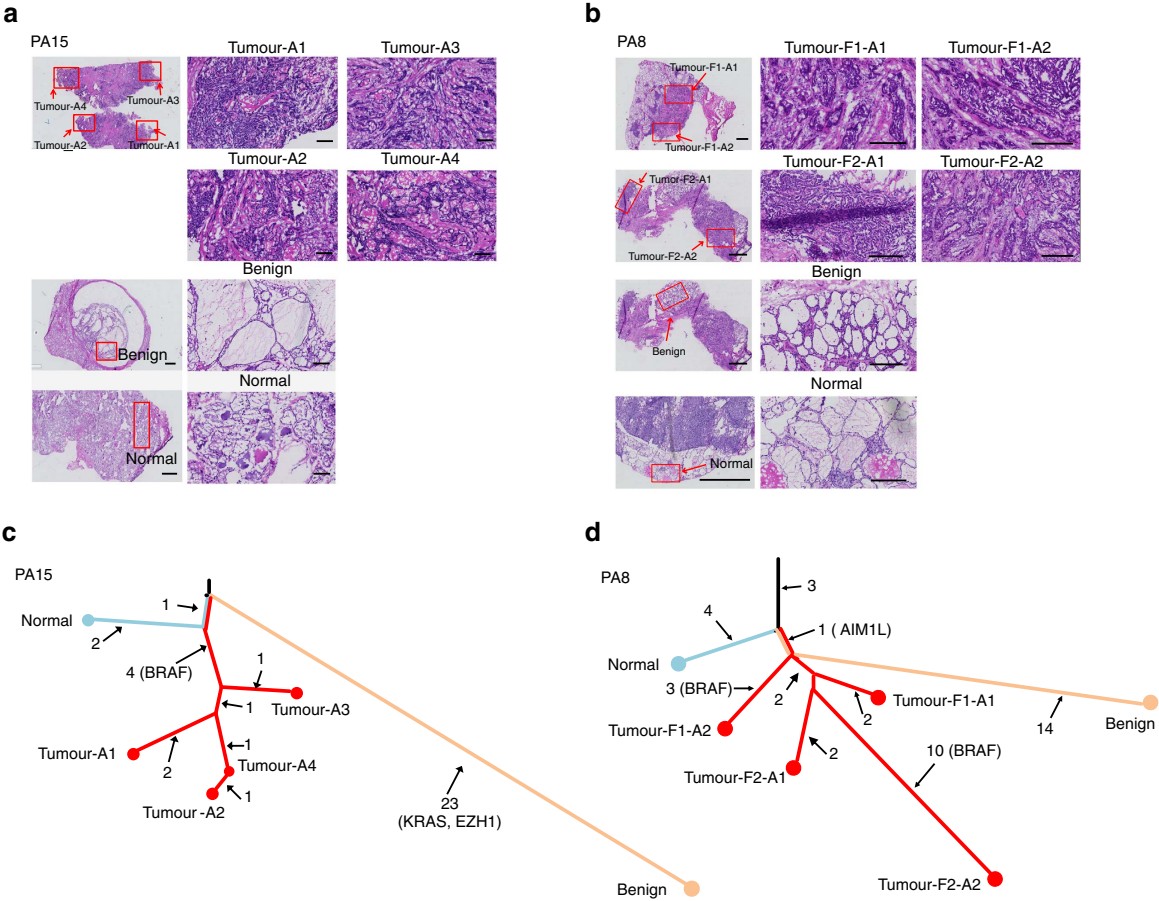

**Figure 5 | Phylogenetic analysis of individual patients demonstrates independent sample origins.** Two representative trees show examples of unrelated (PA15) and distantly related (PA8). (**a,b**) H&E staining of sampled thyroid tissues from patient PA15 (**a**) and PA8 (**b**). The left panel provides a histological view of the entire sample (bar, 1 cm) with the sampling region indicated by red box. Enlarged views of the sampled regions are shown in the right panels (bar, 200 μm). (**c,d**) Phylogenetic trees based on mutation observed in PTC tumour, benign nodule(s) and normal tissue. The length of each line is proportional to the quantity of somatic mutations. The number of mutated genes is indicated near the line, and genes with key non-synonymous mutations were listed.

**RNA sequencing.** Total RNA was prepared with QIAzol Lysis Reagent and RNeasy MinElute Cleanup Kit (Qiagen, Hilden, Germany). On the basis of the resulting RNA integrity analysis, samples were divided into two sets for transcriptome sequencing. Set 2 (RNA integrity number, RIN > 7.0) included tissue samples from two TB (PA7 and PA12) and two SB patients (PB5 and PB6). The cDNA library construction for these samples was performed using Illumina TruSeq RNA Sample Preparation Kit. Set 1 (RIN > 6.0) included four TB patients (PA3–PA6) and four SB patients (PB1–PB4) from the WES cohort, and an independent set PA21. The cDNA library was constructed using Ribo-Zero Gold Kit (Illumina, San Diago, CA). A total of 40 samples (Supplementary Data 6) were sequenced using Illumina HiSeq2500 platform, and 100-bp paired-end reads were generated.

**RNA-Seq data analysis.** The RNA-Seq reads were mapped to the human genome (hg19) and transcriptome (gencode version 19) using the software RNA STAR (version 2.4), which can align reads across splice junctions with or without gene annotations. Gene expression was calculated as FPKM with R package edgeR (version 3.8.5). Genes with FPKM values less than 1 in all samples were removed. Hierarchical clustering analyses were performed using log2 transformation of the FPKM matrix (the FPKM value were adjusted by 0.1 to avoid a logarithm of 0), and 1-Pearson correlation coefficient was used as distances.

We used an R package DEseq2 (version 1.6.1) to compare expression levels between sample pairs. Paired tests were performed, where different patients were treated as biological replicates. We used fold change > 1.5 and FDR < 0.01 as thresholds to define DEGs. The thyroid differentiation and signalling features were evaluated by three groups of genes (16 genes for thyroid differential, 52 genes for ERK activity and 71 for characters of BRAF$^{V600E}$/RAS mutation) as previously reported[3,8]. Briefly, to get TDS score, the median of FPKM values of the 16 genes for thyroid differential (log2-normalized) were calculated, minus the corresponding median and summed across the 16 genes for each sample. The Z-score of FPKM values for the 52 ERK-related genes was used as ERK-activity score. The expression levels of 71 BRAF$^{V600E}$/RAS related genes in each sample in our cohort were used

for hierarchical clustering analyses to display the cancer-related signalling characters of different histological samples.

**Cell proliferation assay.** Nthy-ori 3-1 (ECACC 90011609) cell line was obtained from Prof. Haixia Guan from The First Affiliated Hospital of China Medical University. They were maintained in 1,640 medium (Gibco, Grand Island, NY, USA) supplemented with 10% FBS. Proliferation assay was evaluated by a water-soluble tetrazolium salt (WST) method using Cell Counting Kit-8 (CCK-8, Dojindo Laboratories, Kumamoto, Japan) according to the manufacturer's instructions. Cells were seeded in a 96-well plate in a total volume of 100 μl medium. Quintuplicate wells were assigned to each condition. Cells were grown under standard condition with or without indicated treatment for 48 h. WST was assayed by adding CCK-8 reagent and the absorbance at 450 nm (OD450) and was measured after 2 h reaction using a microplate reader. The relative absorbance of mutation group was normalized to the control group. Means ± s.d. of each quintuplicate was presented in the graph. All experiments were repeated at least three times.

**Cell invasion assay.** Cell invasion experiments were carried out with the QCM 24-well Fluorimetric Cell Invasion Assay kit (ECM554; Chemicon) according to the manufacturer's instructions. Each insert contains an 8 μm pore size poly-carbonate membrane coated with a thin layer of ECMatrixTM. The ECM layer occludes the membrane pores, blocking non-invasive cells from migrating through. Invasive cells, on the other hand, migrate through the ECM layer and cling to the bottom of the polycarbonate membrane. Invaded cells on the bottom of the insert membrane are dissociated from the membrane when incubated with Cell Detachment Buffer and subsequently lysed and detected by CyQuant GR@ dye.

**Data availability.** The authors declare that all data supporting the findings of this study are available within the article and its Supplementary Information files or

from the corresponding author upon reasonable request. The sequence reported in this paper has been deposited in the European Genome-Phenome Archive (EGA) database (accession no. EGAS00001002312).

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

## Acknowledgements

This study was supported by the grants from National Science Foundation of China (81,570,702, 81,402,212, 81,400,772, 81,500,604, 81,130,016 and 81,270,872). National International Science Cooperation Foundation (2015DFA30560).

## Author contributions

W.W., S.W. and G.N. managed and supervised this project. L.Y., X.Z., F.H., L.S., X.F., J.X. and M.C. prepared the samples. L.Y., X.Z., H.X. and Y.S. performed the bioinformatics analysis. F.H. and W.W. performed the validation of mutations. W.W. and Y.Q. performed the functional assay. X.F. and J.X. performed the histological analysis. L.Y., Y.Z. and W.Z. managed the patients. L.Y. and X.Z. wrote the manuscript; W.W., S.W., G.N. and H.X. revised the manuscript.

## Additional information

**Competing interests:** The authors declare no competing financial interests.

**DOI: 10.1038/ncomms16129**   **OPEN**

# Corrigendum: The genetic landscape of benign thyroid nodules revealed by whole exome and transcriptome sequencing

Lei Ye, Xiaoyi Zhou, Fengjiao Huang, Weixi Wang, Yicheng Qi, Heng Xu, Yang Shu, Liyun Shen, Xiaochun Fei, Jing Xie, Min Cao, Yulin Zhou, Wei Zhu, Shu Wang, Guang Ning & Weiqing Wang

*Nature Communications* 8:15533 doi: 10.1038/ncomms15533 (2017); Published 5 Jun 2017; Updated 4 Jul 2017

The original version of this Article contained an error in the formatting of the author name Yang Shu, which was incorrectly given as Shu Yang. This has now been corrected in both the PDF and HTML versions of the Article.

