## [Peer Review File · Nature Communications]

Reviewers' Comments:

Reviewer #1 (Remarks to the Author)

Ye and colleagues utilize 21 patients and analyze DNA and RNA differences between their PTC's and accompanying benign nodules. They found that somatic mutation of BRAF (22/32) was only detected in PTC, while mutations in SPOP 39 (4/38), ZNF148 (6/38) and EZH1 (3/38) were found enriched in benign nodules. They conclude (and their title reflects this) that PTC and benign thyroid nodules have genetically independent origins.

1. For the authors to make their conclusion, they are assuming that there was a question that benign thyroid nodules and PTC shared genetic origin or that PTC evolved from benign nodules. While this was a question over 15 years ago, and the question was whether follicular adenomas or hyperplasias evolved to PTC, this question has been settled a long time ago: we know that benign thyroid nodules and PTC have independent origins and hence, what the authors are addressing is a non-question in 2016.

2. Having said that, could the authors only present their data, especially on the expanded series of benign nodules? I would argue yes because genomic alterations in thyroid benign masses have been ill-studied although the PTC genomic data are superfluous in view of TCGA and many other publications preceding. I would suggest the authors remove any form of speculation about thyroid carcinogenesis and from making clinical implications (last sentence of their manuscript, eg) because they are either non-questions or they are just plain incorrect, and present this as a fundamental basic knowledge manuscript.

3. The authors also keep referring to benign thyroid nodules. There are in fact many different histologies. What have they used as this may affect the somatic alterations therefrom?

4. If we assume this is a fundamental knowledge type of manuscript, then the authors should functionally interrogate some of the variants they identify in the benign thyroid nodules.

5. Have the authors looked for large deletions/rearrangements (eg with XHMM)?

Reviewer #2 (Remarks to the Author)

This is an exciting study that for the first time reveal the genetic landscape of benign thyroid neoplasm by whole exome sequencing and transcriptome sequencing. The authors have discovered several unique mutations in this common thyroid neoplasm, including mutations ZNF1 (21.4%), SPOP (14.3%), and EZH1 (10.7%) genes, which were not found in PTC or normal thyroid tissues. These mutations were found to be mutually exclusive, suggesting they may each play an important role in the tumorigenesis of benign thyroid neoplasm. The main known genetic alteration in benign thyroid neoplasms is sometimes RAS mutation. The present study provides an important new dimension to the genetic background that accounts for a large number of benign thyroid neoplasms. Given the specificity of these mutations for the benign thyroid nodules demonstrated in this study, they are valuable diagnostic molecular markers that can be used to rule out malignancy. This will likely be very helpful particularly in the diagnostic evaluation of thyroid nodules that are cytologically indeterminate—a common diagnostic dilemma encountered clinically. This study thus has important genetically mechanistic and clinical implications for benign thyroid neoplasm, a very common clinical condition.

The study is overall well performed and the analyses are comprehensive. There are a few minor issues as stated below that may be addressed/discussed to improve the manuscript.

1) Given that this study proves no genetic link between benign thyroid nodule (probably "neoplasm" more accurate, see below) and papillary thyroid cancer (PTC), the authors conclude that PTC does not derive from benign nodules. It is in fact conventionally believed that follicular thyroid cancer (FTC), not PTC, can derive from benign thyroid nodules (e.g., Xing, Nature Reviews Cancer 2013, 13:184-99). It is advisable that the authors state that their findings are actually consistent with the conventional belief.

2) The suggestion by the authors that the finding of "genetic separation" between benign thyroid nodule and PTC may help prevent overtreatment of thyroid nodule is interesting. But this will likely be the case if future studies show that the mutations found in the present study are specific for benign thyroid nodule, but not seen in FTC. Because a challenging diagnostic dilemma currently encountered in the clinic is the difficulty differentiating benign thyroid nodule from FTC, not much PTC. If the new mutations found in the present study are specific for benign thyroid nodule, they will help identify these nodules to spare them from surgery. This is worth a brief discussion in the manuscript.

3) By the term "goiter", the authors probably meant "hyperplasia" in the manuscript (the conventional definition for "goiter" is simply "enlarged thyroid gland" regardless of the etiology). It would be interesting to see how the mutations discovered here are distributed among hyperplasia and neoplasm (i.g., adenoma). It is likely that the mutation prevalence is much higher in the latter or perhaps exclusively in the latter since the latter is neoplasms (tumor).

4) The authors may want to consider the use of "benign thyroid neoplasm" (or "adenoma") to replace "thyroid nodule" if the above #3 proves to be the case. The term "Thyroid nodule" is a vague and non-specific term that does not tell the nature of the condition; for example, a thyroid nodule can be an inflammatory lesion (as seen in Hashimoto's thyroiditis), a hyperplasia lesion (not neoplasm, i.e., not tumor), etc.

5) Were any RAS mutations found?

Again, this is an important novel study that adds substantially to our current understanding of the genetic backgrounds of benign thyroid neoplasm, with strong biological and clinical implications. (Mingzhao Xing).

Reviewer #3 (Remarks to the Author)

Ye et al performed a timely and well-designed study to identify genetic signals that lead to benign thyroid nodules. They carefully compared mutational landscapes of normal - benign nodules - thyroid cancer and concluded that benign nodules and thyroid cancers are genetically distinct, which led to the genetic evidence that thyroid nodule surgery has been over-used. Data are high quality, the conclusion is quite solid and the topic is of high interest, but I have some concerns over interpretation of the results that can be addressed.

Major concerns:

1. The authors compared PTC profile (~2/3 with BRAF mutation) with benign nodule profile (no BRAF, instead ~1/3 SPOP, ZNF148 and EZH1 mutations) and concluded that benign nodules won't evolve into PTC. It is a reasonable conclusion given that SPOP, ZNF148 and EZH1 mutations are rare in PTC. However, what would be the odd of ~2/3 of benign nodules without the SPOP/ZNF148/EZH1 mutations evolve into PTC? In other words, ~1/3 of nodules with S/Z/E mutations will remain as benign, but ~2/3 of nodules without S/Z/E mutations still possess potential to acquire BRAF mutation and become a cancer. On the same line, how would you defend if the presence of SPOP/ZNF148/EZH1 mutations would cause nodules, but also somehow suppress

the tissue progress into the cancerous status?

2. Variant allele fraction of mutations from normal, benign nodules, and PTC should be compared and their purity should be taken into account when interpreting the variant call result and tumor evolution.

3. SPOP in prostate, ZNF148 and EZH1 in uveal melanoma, clinical characteristics of the tumors, additional drivers co-exist that distinct from benign nodules

4. The study does not touch any of the structural variations (somatic CNV, LOH, translocation etc)? How does the SV burden look like between normal - nodules - PTC? Is there any specific signal that are distinct between benign nodules and PTC?

Minor concerns:

1. Fig 3. Does PA10 B-F1 have 0 mutation? And PA10 B-F2, PA6 B-F3 and PA18 B appear to have very low number of mutations. I wonder if it is fair to compare tumor vs benign module if absolute numbers are low - that will obviously causes low overlap fraction. Therefore, can you perform similar comparison with randomly selected T vs T or B vs B? (Fig S5 contains such comparison but does not factor in mutation numbers)

2. Fig 4 c, d are eyesores. Should be visually improved.

3. Fig 5c, unsupervised clustering of DEGs sounds more like supervised clustering. Performing PCA using entire gene set to test if normal - benign nodules - PTC display separate clusters would provide stronger evidence that they form genetically different groups.

Reviewer #1 (Remarks to the Author):

Ye and colleagues utilize 21 patients and analyze DNA and RNA differences between their PTC's and accompanying benign nodules. They found that somatic mutation of BRAF (22/32) was only detected in PTC, while mutations in SPOP 39 (4/38), ZNF148 (6/38) and EZH1 (3/38) were found enriched in benign nodules. They conclude (and their title reflects this) that PTC and benign thyroid nodules have genetically independent origins.

1. For the authors to make their conclusion, they are assuming that there was a question that benign thyroid nodules and PTC shared genetic origin or that PTC evolved from benign nodules. While this was a question over 15 years ago, and the question was whether follicular adenomas or hyperplasias evolved to PTC, this question has been settled a long time ago: we know that benign thyroid nodules and PTC have independent origins and hence, what the authors are addressing is a non-question in 2016.

Thank you for your comments. We totally agree with you. The focus of the current study should be the genetic study of benign thyroid nodule. As you mentioned, unlike PTC which has been extensively studied, benign thyroid nodule is ill-studied. We are the first to illustrate the genetic landscape of benign thyroid nodule. We found benign thyroid nodules have a genetic and transcriptome landscape distinct from PTC tumors. Mutually exclusive *SPOP*^{P94R}, *EZH1*^{Q571R}, and *ZNF148* mutations were identified in 24.3% of benign nodules.

Also as you said, for many years, we believe PTC does not derive from benign thyroid nodules, mainly from epidemiology data and molecular study, e.g BRAF V600E in a significant subset of PTC while is lacking in benign nodule. In addition to these, more genomic evidence would further support this belief. By exome and transcriptome study of the benign nodules, we illustrated the genetic features of benign nodules, which are distinct from the coincidental PTC tumors, thus providing solid genetic evidence for the conventional belief.

You have raised a very good point. Focusing on benign nodule would make the present study more straightforward. Therefore, we have reframed the manuscript focusing on benign nodules and removed all the speculation about thyroid carcinogenesis as well as the clinical implications. Thank you.

2. Having said that, could the authors only present their data, especially on the expanded series of benign nodules? I would argue yes because genomic alterations in thyroid benign masses have been ill-studied although the PTC genomic data are superfluous in view of TCGA and many other publications preceding. I would suggest the authors remove any form of speculation about thyroid carcinogenesis and from making clinical implications (last sentence of their manuscript, eg) because they are either non-questions or they are just plain incorrect, and present this as a fundamental basic knowledge manuscript.

We are really appreciated with your suggestions. Defining the genomic alterations associated with benign thyroid nodule should be the primary goal of the present study. By doing that, the flow of our manuscript would be more smooth and straightforward. According to your suggestion, we have reframed the manuscript focusing on benign nodules, addressing benign nodule related genetic alterations in a large sample set, the mutant frequency in TCGA database, the functional role of the mutant allele and the comparison with PTC. We have removed all the speculation about thyroid carcinogenesis as well as the clinical implications:

1) Statement “their potential genetic contributions to tumorigenesis of thyroid cancer” and “Furthermore, the early genetic separation of PTC from benign nodule suggests the PTCs do not derive from benign nodules providing molecular evidence that could be used to prevent over-treatment of patient with thyroid nodules.” in the abstract was deleted.

2) Statement “Concerns related to the potential malignant transformation from benign thyroid nodule (benign nodule) to thyroid carcinoma continue to plague both patients and medical professionals, thus leading to over-management.” in the first paragraph of the main text was deleted.

3) Statement “Accumulated clinic evidence argues against the direct transformation of benign nodules to thyroid cancer, despite that some benign polyps are believed to have malignant transformation potential (e.g., intestinal polyps), thus recommended to be surgically removed. This approach is supported by genetic evolution studies examining benign precursor lesions through malignancy (e.g., shared BRAFV600E mutation in nevus and melanoma). Genetically, key driver mutations identified in thyroid cancer are rarely detected in benign nodules, with only a small fraction of benign nodules found to harbor somatic mutations of the RAS, RET-PTC rearrangement, or EIF1AX mutation. However, the persistent concern that benign nodules serve as the precursor lesions of thyroid cancer has caused huge medical and mental burden.” in the second paragraph of the discussion was deleted.

4) Statement “These findings support that the origin and evolutionary pathway to benign nodule development is distinct from PTC tumor formation. Thus, routine surveillance of benign thyroid nodules may not be necessary, especially where SPOP, ZNF148, and EZH1 mutations have been detected. Given that nearly 80% of nodule biopsies are clearly defined as benign, with another 15% labeled as indeterminate, the potential clinical impact is significant.” in the last paragraph was deleted.

3. The authors also keep referring to benign thyroid nodules. There are in fact many different histologies. What have they used as this may affect the somatic alterations therefrom?

Thank you very much for your comments. You have raised a very important point. The histology of the benign thyroid nodule is adenomatoid nodule in our study. As you said, benign thyroid nodule included many histological types. According to Diagnostic Pathology Endocrine edited by Vania Nose (first published in 2012, ISBN:978-1-931884-59-4) and Endocrine Pathology edited by Lester DR Thompson (first published in 2006, ISBN-13:978-0-443-06685-6), there are nonneoplastic nodule such as inflammatory nodule caused by acute or Hashimoto thyroiditis and adenomatoid nodule; benign neoplasm such as follicular adenoma. Different pattern of somatic alterations may be associated with different histology. Mutations in TSHR, GNAS and *EZH1*^{Q571R} have been found in autonomous thyroid adenomas (J Clin Invest. 2016 doi:10.1172/JCI84894). The current study focused on adenomatoid nodule, including 5 cellular subtype in the discovery set (Supplementary Table 1, 2) and 35 cellular subtype in the validation set. The others were classic subtype. We found recurrent mutations of *SPOP*^{P94R} and *EZH1*^{Q571R}, and in the last exon of *ZNF148*. Your comments remind us to specify the expression of “adenomatoid nodule” to distinguish from other types of benign nodule in the revised manuscript. Thank you again.

4. If we assume this is a fundamental knowledge type of manuscript, then the authors should functionally interrogate some of the variants they identify in the benign thyroid nodules.

Thank you for your comments. We have performed functional analysis of the three genes with the most frequent mutation in normal human thyroid cell line (Nthy). Because hotspot mutations were found in *SPOP* and *EZH1* gene, we evaluated cell proliferation and cell invasion after overexpression of *SPOP*^{P94R} and *EZH1*^{Q571R}. A total of 12 mutations were found in *ZNF148*, all located in the last exon, we therefore evaluated cell proliferation and cell invasion after knock down of *ZNF148* by siRNA. Only slightly increased cell proliferation ($p=0.04$) and reduced cell invasion ($p=0.04$) was observed after overexpression of *EZH1*^{Q571R} compared to WT. No significant change of cell proliferation or invasion was observed after overexpression of *SPOP*^{P94R} or knock down of *ZNF148* by siRNA. (Line 18-20, Page 4, Supplementary Fig.5)

According to your suggestions, we performed the above experiments on one more cell line (293T) to confirm this, and there was only slight reduced cell invasion ($p=0.04$) but no significant proliferation change after knock down of *ZNF148* by siRNA. No significant change was observed for either proliferation or invasion assay, after overexpression of *SPOP*^{P94R} or *EZH1*^{Q571R}. These data excluded their role of inducing thyroid carcinogenesis, however their role in benign nodule formation is still waiting for deeper investigations in future.

5. Have the authors looked for large deletions/rearrangements (eg with XHMM)?

Thank you for your suggestion. We analyzed copy number variation (CNV) using the exome sequencing data and rearrangements using the transcriptome sequencing data in the revised version (Line 18-20, Page 4, Supplementary Fig.6 and 7). We found no recurrent arm-level CNV in benign samples, although a previous reported PTC related alternation, 22q-del, was identified in 9 PTC samples (from 7 patients) (Supplementary Fig. 6). As the limitations of whole exome sequence data for CNV analysis, we only focus the arm level CNVs but not focal CNVs. We did not detect any recurrent or functional transcript fusion in either benign nodules or PTC (Supplementary Fig.7).

Reviewer #2 (Remarks to the Author):

This is an exciting study that for the first time reveal the genetic landscape of benign thyroid neoplasm by whole exome sequencing and transcriptome sequencing. The authors have discovered several unique mutations in this common thyroid neoplasm, including mutations ZNF1 (21.4%), SPOP (14.3%), and EZH1 (10.7%) genes, which were not found in PTC or normal thyroid tissues. These mutations were found to be mutually exclusive, suggesting they may each play an important role in the tumorigenesis of benign thyroid neoplasm. The main known genetic alteration in benign thyroid neoplasms is sometimes RAS mutation. The present study provides an important new dimension to the genetic background that accounts for a large number of benign thyroid neoplasms. Given the specificity of these mutations for the benign thyroid nodules demonstrated in this study, they are valuable diagnostic molecular markers that can be used to rule out malignancy. This will likely be very helpful particularly in the diagnostic evaluation of thyroid nodules that are cytologically indeterminate—a common diagnostic dilemma encountered clinically. This study thus has important genetically mechanistic and clinical implications for benign thyroid neoplasm, a very common clinical condition.

The study is overall well performed and the analyses are comprehensive. There are a few minor issues as stated below that may be addressed/discussed to improve the manuscript.

1) Given that this study proves no genetic link between benign thyroid nodule (probably “neoplasm” more accurate, see below) and papillary thyroid cancer (PTC), the authors conclude that PTC does not derive from benign nodules. It is in fact conventionally believed that follicular thyroid cancer (FTC), not PTC, can derive from benign thyroid nodules (e.g., Xing, Nature Reviews Cancer 2013, 13:184-99). It is advisable that the authors state that their findings are actually consistent with the conventional belief.

We are very appreciated with your suggestion. In the revised manuscript, we have stated that our findings are actually consistent with the conventional belief in Line 18-19 in the abstract, Line 16-18 Page 3 and Line 14-15, 27-28 Page 8. The previous statement that PTCs do not derive from benign nodules has been deleted. To illustrate the model of the progression of thyroid tumorigenesis, we added one more reference (Molecular pathogenesis and mechanisms of thyroid cancer. Nat Rev Cancer. 2013 Mar;13(3):184-99.) in the revised manuscript as REF 7.

2) The suggestion by the authors that the finding of “genetic separation” between benign thyroid nodule and PTC may help prevent overtreatment of thyroid nodule is interesting. But this will likely be the case if future studies show that the mutations found in the present study are specific for benign thyroid nodule, but not seen in FTC. Because a challenging diagnostic dilemma currently encountered in the clinic is the difficulty differentiating benign thyroid nodule from FTC, not much PTC. If the new mutations found in the present study are specific for benign thyroid nodule, they will help identify these nodules to spare them from surgery. This is worth a brief discussion in the manuscript.

We are really appreciated with your suggestion. You have raised a very important and interesting point. We followed your suggestions and sequenced *SPOP*^{P94R} and *EZH1*^{Q571R}, and the last exon of ZNF148 in a small set of FTC samples (N=55) by PCR-Sanger sequencing. You are absolutely right; no mutation was found in FTC. Consistently, in a recent published study which included 13 FTCs, exome sequencing did not detect *SPOP*, *EZH1* or ZNF148 mutation either (Oncotarget, 2016). Nevertheless, these findings should be validated in future study with large sample size and follow-up data.

We therefore added the following discussion in Line 27-28 Page 7 and Line 1-5 Page 8. Thank you again.

“A challenging diagnostic dilemma currently encountered in the clinic is the difficulty differentiating benign follicular lesion from FTC. We sequenced *SPOP*^{P94R} and *EZH1*^{Q571R}, and the last exon of ZNF148 in an additional set of 55 follicular thyroid carcinomas (FTC). No mutation was found in FTC. These data suggest these mutations are specific for benign nodule and identify nodules with these mutations may spare them from surgery when follicular lesion of undetermined significance or a follicular neoplasm was suspected in fine-needle aspiration.”

3) By the term “goiter”, the authors probably meant “hyperplasia” in the manuscript (the conventional definition for “goiter” is simply “enlarged thyroid gland” regardless of the etiology). It would be interesting to see how the mutations discovered here are distributed among hyperplasia and neoplasm (i.g., adenoma). It is likely that the mutation prevalence is much higher in the latter or perhaps exclusively in the latter since the latter is neoplasms (tumor).

Thank you for remind us again. Yes, by “goiter” we mean hyperplasia. You have raised a very interesting question of how the mutations discovered are distributed among hyperplasia (e.g adenomatoid nodule) and benign neoplasm (e.g follicular adenomas/FA). Following your suggestion, we sequenced *SPOP*^{P94R} and *EZH1*^{Q571R}, and the last exon of ZNF148 in an additional set of 17 FAs by PCR-Sanger sequencing. The result is shown in the following table. We also included data from the recent published study (Oncotarget, 2016) in the table. As you speculated, mutations especially *EZH1*^{Q571R}, were also identified in FA, but with inconsistent prevalence. *EZH1*^{Q571R} was identified in 4% of FA based upon our data, while in 21.4% (3/14) of FA according to the recent study (Oncotarget, 2016). This discrepancy may be caused by different tissue type and sequencing platform.

	Adenomatoid nodule (N=259)	FA (N=17)	FA (N=14)
Data origin	The present study	Our work	Oncotarget, 2016
Tissue type	Frozen	FFPA	Frozen
Sequencing platform	Exome sequencing with Sanger validation (N=28) and PCR-Sanger (N=231)	PCR-Sanger	Exome sequencing without Sanger validation
SPOP ^{P94R}	11.2%	0	7.1%
EZH1 ^{Q571R}	9.3%	5.9%	21.4%
ZNF148	5.4%	0	0

4) The authors may want to consider the use of “benign thyroid neoplasm” (or “adenoma”) to replace “thyroid nodule” if the above #3 proves to be the case. The term “Thyroid nodule” is a vague and non-specific term that does not tell the nature of the condition; for example, a thyroid nodule can be an inflammatory lesion (as seen in Hashimoto’s thyroiditis), a hyperplasia lesion (not neoplasm, i.e., not tumor), etc.

Again, thank you so much for pointing this out so patiently. The benign nodules in the present study are hyperplasia lesions and specifically are adenomatoid nodules according to Diagnostic Pathology Endocrine edited by Vania Nose (first published in 2012, ISBN:978-1-931884-59-4) and Endocrine Pathology edited by Lester DR Thompson (first published in 2006, ISBN-13:978-0-443-06685-6). Five nodules in the discovery set and 35 in the validation set were cellular adenomatoid nodules, the others are classic adenomatoid nodule (Supplementary Table 1, 2). Classic adenomatoid nodules are featured as enlarged follicles distended with colloid, lined by flattened follicular epithelial cells. Cellular adenomatoid nodule appears solid, with minimal colloid and is difficult to distinguish with follicular adenoma. However, all our patients had two or more nodules and the nodules were without capsule, thus excluding follicular adenoma. We therefore followed your suggestion, and specified “benign thyroid nodule” as “adenomatoid nodules” in the revised manuscript. Thank you again.

5) Were any RAS mutations found?

Yes, we found RAS mutation in 3/28 benign nodules in the discovery dataset, including 2 NRAS mutation and 1 KRAS mutation. There is no RAS mutation in the coincidental PTCs. We did not sequence RAS in the validation samples.

Again, this is an important novel study that adds substantially to our current understanding of the genetic backgrounds of benign thyroid neoplasm, with strong biological and clinical implications. (Mingzhao Xing).

Reviewer #3 (Remarks to the Author):

Ye et al performed a timely and well-designed study to identify genetic signals that lead to benign thyroid nodules. They carefully compared mutational landscapes of normal - benign nodules - thyroid cancer and concluded that benign nodules and thyroid cancers are genetically distinct, which led to the genetic evidence that thyroid nodule surgery has been over-used. Data are high quality, the conclusion is quite solid and the topic is of high interest, but I have some concerns over interpretation of the results that can be addressed.

Major concerns:

1. The authors compared PTC profile (~2/3 with BRAF mutation) with benign nodule profile (no BRAF, instead ~1/3 SPOP, ZNF148 and EZH1 mutations) and concluded that benign nodules

won't evolve into PTC. It is a reasonable conclusion given that SPOP, ZNF148 and EZH1 mutations are rare in PTC. However, what would be the odd of ~2/3 of benign nodules without the SPOP/ZNF148/EZH1 mutations evolve into PTC? In other words, ~1/3 of nodules with S/Z/E mutations will remain as benign, but ~2/3 of nodules without S/Z/E mutations still possess potential to acquire BRAF mutation and become a cancer. On the same line, how would you defend if the presence of SPOP/ZNF148/EZH1 mutations would cause nodules, but also somehow suppress the tissue progress into the cancerous status?

Thank you for your comments. You are quite right. The ~2/3 of nodules without S/Z/E mutations still possess potential to acquire BRAF mutation and become a cancer. This is a very important question but is out of the scope of the current study. We will work on this in the future study. As to the second question, our data only showed the distinct genomic alterations between benign nodule and thyroid cancer. To defend if the presence of SPOP/ZNF148/EZH1 mutations would cause nodules, but also somehow suppress the tissue progress into the cancerous status, we would have to see if these mutations could suppress e.g BRAF induced thyroid carcinogenesis, which is out of the scope of the current study. Thank you for bringing this point, we will also work on it in future work.

2. Variant allele fraction of mutations from normal, benign nodules, and PTC should be compared and their purity should be taken into account when interpreting the variant call result and tumor evolution.

Thank you for your comments. In fact, the correlations of paired samples (normal, benign nodules and PTC) were based on the variant allele fraction (VAF) of mutations (Supplementary Fig. 10), and the phylogenetic trees were constructed based on these correlations. We rewrote the method section and displayed the comparisons of VAF of all mutations called from paired samples in each patient (Line 6-12, Page 5, Supplementary Fig. 9 and 14). Furthermore, we inferred the purity and ploidy by using ABSOLUTE (Sup. Table 8), and then estimated their effects on mutation calling (Line 6-12, Page 5) according to the method from a previous study (Nature Genet. 2015, 47(9):1038-1046). There was no significant correlation between mutation number and the purity (Spearman's rho, $r=-0.037$, $p=0.76$), and the purities of paired samples was not significant different ($p=0.443$, paired t-test), both of which suggested that mutation calling and evolution analysis was affected by the purity.

3. SPOP in prostate, ZNF148 and EZH1 in uveal melanoma, clinical characteristics of the tumors, additional drivers co-exist that distinct from benign nodules

Thank you so much for your comments. You have made a great point. Although mutations of SPOP, ZNF148 and EZH1 have been found in malignancies like prostatic cancer and uveal melanoma, they tended to have co-existence of additional driver alterations in well-known cancer-related genes (e.g., BRAF, MLL2). Benign thyroid nodule on the other hand, based upon our study, does not have additional drive mutations, which make them benign rather than malignancy.

Thank you. We added the following descriptions in our discussion part in Line 11-17 Page 7.

“SPOP mutations have been found in TCGA prostate cancer (10.2%), but with ~70% has altered either Tryptophan residue at 131 position or Phenylalanine residue at 133 position (Supplementary Fig. 4), while mutations of ZNF148 and EZH1 were commonly observed in melanoma as well with no hot mutation spot. However, patients tended to have co-existence of alterations in well-known cancer-related genes (e.g., BRAF, MLL2) in these malignant tumors, which is distinct from benign nodules.”

4. The study does not touch any of the structural variations (somatic CNV, LOH, translocation etc)? How does the SV burden look like between normal - nodules - PTC? Is there any specific signal that are distinct between benign nodules and PTC?

We totally agree with your idea that information of more type of variations would be helpful to reveal the accurate relationship benign nodules and PTC. We analyzed the arm-level CNVs and rearrangements and the results have been added in the revised version (Line 20-25, Page 4, Supplementary Fig. 6 and 7). Briefly, we found a previously reported PTC related CNV, 22q-loss, in our PTC cohort, but not in any benign samples, implying that the CNVs were also different in benign nodule and PTC.

Your suggestions help us find another support that PTC and benign nodules are different. It's a pity that, due to the limitations of the whole exome sequencing data was not appropriate for more comprehensive SV analyses and future whole genome sequencing would be helpful. And thanks again for your suggestion.

Minor concerns:

1. Fig 3. Does PA10 B-F1 have 0 mutation? And PA10 B-F2, PA6 B-F3 and PA18 B appear to have very low number of mutations. I wonder if it is fair to compare tumor vs benign module if absolute numbers are low - that will obviously causes low overlap fraction. Therefore, can you perform similar comparison with randomly selected T vs T or B vs B? (Fig S5 contains such comparison but does not factor in mutation numbers)

T Thank you for your comments. No nonsynonymous mutation was called in PA10-B-F1, and some samples have only a few mutations. In fact, the low mutation level resulted in a relative high overlap fraction as the denominator is low. As the genetic variation in individual patients is distinct, T-T or B-B in different patient is not comparable. This was why we used paired samples in the same patient to demonstrate the genetic relationship between benign nodule and thyroid cancer. As you suggested, we analyzed the relationship between mutation number and overlap ratio (Supplementary Fig. 8c and 8d), which showed that the overlap ratio and mutation number are not correlated ($p=0.56$).

2. Fig 4 c, d are eyesores. Should be visually improved.

Thank you for your comments. We have modified Fig. 4c,d, which is currently Fig. 5c,d.

3. Fig 5c, unsupervised clustering of DEGs sounds more like supervised clustering. Performing PCA using entire gene set to test if normal - benign nodules - PTC display separate clusters would provide stronger evidence that they form genetically different groups.

Thanks for your comments. We used all the expressed transcripts (21,441 with the FPKM value \geq 1 in at least one sample) to perform PCA (Line 25, Page 5, Supplementary Fig. 11), and did find normal - benign nodules - PTC display separate clusters.

Reviewers' Comments:

Reviewer #1 (Remarks to the Author)

Thank you for taking up all my suggestions. I am concerned about the minute functional effects of your identified mutations. Please could you flush out the function a bit more. Right now, the functional data virtually negates your beautiful genomic analyses. Could you look at invasion (Boyden chamber, not just scratch test), EMT, apoptosis, cell cycle and changes in stem-ness?

Reviewer #2 (Remarks to the Author)

The authors took much effort to thoroughly address my critiques, which is satisfactory. A particular effort of the authors that should be commended is that they followed my suggestion to sequence the major mutations, such as the SPOPP94R and EZH1Q571R and the last exon of ZNF148, in a cohort (55 cases) of FTC and found no mutation in the latter. This is an exciting result as it suggests that these mutations are benign thyroid nodule (benign neoplasm)-specific and can therefore be very useful diagnostic molecular markers that can help rule out FTC which is often cytologically indistinguishable on needle biopsy. This represents a significant advancement in the genetic molecular marker field of thyroid tumor medicine. It is also reassuring that these findings in the present study have been confirmed in a recent paper published during the review of this manuscript on a smaller-scale study (Oncotarget, 2016).

Reviewer #3 (Remarks to the Author)

Most of the comments are well addressed. I wish to see the follow up studies that can directly interrogate genetic relationships during different developmental stages of thyroid tumors in the near future.

REVIEWERS' COMMENTS:

Reviewer #1 (Remarks to the Author):

Thank you for taking up all my suggestions. I am concerned about the minute functional effects of your identified mutations. Please could you flush out the function a bit more. Right now, the functional data virtually negates your beautiful genomic analyses. Could you look at invasion (Boyden chamber, not just scratch test), EMT, apoptosis, cell cycle and changes in stem-ness?

Thank you for your comments. We have discussed the limitations of our functional data as the following in the revised version: "However, we did not investigate the capacity of thyroid hormone synthesis and transport, which may be compromised in thyroid nodules with somatic SPOP, ZNF148 and EZH1 mutation, causing enlarged follicles distended with colloid as featured in adenomatoid nodule. Further studies are needed." at Line 8 Page 8.

Reviewer #2 (Remarks to the Author):

The authors took much effort to thoroughly address my critiques, which is satisfactory. A particular effort of the authors that should be commended is that they followed my suggestion to sequence the major mutations, such as the SPOPP94R and EZH1Q571R and the last exon of ZNF148, in a cohort (55 cases) of FTC and found no mutation in the latter. This is an exciting result as it suggests that these mutations are benign thyroid nodule (benign neoplasm)-specific and can therefore be very useful diagnostic molecular markers that can help rule out FTC which is often cytologically indistinguishable on needle biopsy. This represents a significant advancement in the genetic molecular marker field of thyroid tumor medicine. It is also reassuring that these findings in the present study have been confirmed in a recent paper published during the review of this manuscript on a smaller-scale study (Oncotarget, 2016).

Thank you for your comments.

Reviewer #3 (Remarks to the Author):

Most of the comments are well addressed. I wish to see the follow up studies that can directly interrogate genetic relationships during different developmental stages of thyroid tumors in the near future.

Thank you for your comments.